# Graph Representation Learning-Based Early Depression Detection Framework in Smart Home Environments

**DOI:** 10.3390/s22041545

**Published:** 2022-02-17

**Authors:** Jongmo Kim, Mye Sohn

**Affiliations:** Department of Industrial Engineering, Sungkyunkwan University, Suwon 16419, Korea; dignityc@skku.edu

**Keywords:** smart home, early detection of depression (EDD), elderly, graph neural networks, graph representation learning, knowledge graph

## Abstract

Although the diagnosis and treatment of depression is a medical field, ICTs and AI technologies are used widely to detect depression earlier in the elderly. These technologies are used to identify behavioral changes in the physical world or sentiment changes in cyberspace, known as symptoms of depression. However, although sentiment and physical changes, which are signs of depression in the elderly, are usually revealed simultaneously, there is no research on them at the same time. To solve the problem, this paper proposes knowledge graph-based cyber–physical view (CPV)-based activity pattern recognition for the early detection of depression, also known as KARE. In the KARE framework, the knowledge graph (KG) plays key roles in providing cross-domain knowledge as well as resolving issues of grammatical and semantic heterogeneity required in order to integrate cyberspace and the physical world. In addition, it can flexibly express the patterns of different activities for each elderly. To achieve this, the KARE framework implements a set of new machine learning techniques. The first is 1D-CNN for attribute representation in relation to learning to connect the attributes of physical and cyber worlds and the KG. The second is the entity alignment with embedding vectors extracted by the CNN and GNN. The third is a graph extraction method to construct the CPV from KG with the graph representation learning and wrapper-based feature selection in the unsupervised manner. The last one is a method of activity-pattern graph representation based on a Gaussian Mixture Model and KL divergence for training the GAT model to detect depression early. To demonstrate the superiority of the KARE framework, we performed the experiments using real-world datasets with five state-of-the-art models in knowledge graph entity alignment.

## 1. Introduction

### 1.1. Background

Depression is the most common psychiatric disorder in adults, accompanied by symptoms such as a depressed mood, lack of motivation, changes in appetite, trouble sleeping, decreased energy, fatigue, fatigue, and anxiety [1,2]. According to the World Health Organization (WHO), in 2021, 3.8% of the world’s population suffered from depression, and the rate in the elderly over the age of 60 was 5.7%, 1.5 times the overall level [3]. It should pay particular attention to depression of the elderly because it is difficult for the elderly to recognize that they are depressed on their own, so they often complain of physical symptoms rather than complaining of changes in their emotions [4,5].

Although diagnosing and treating depression is a medical field, rapidly developing information and communication technologies (ICTs) and artificial intelligence (AI) technologies contribute to timely diagnosis and treatment through its early detection [6,7,8,9,10,11,12]. Early Detection of Depression (EDD) research using ICTs and AI technology is divided into two aspects. The first one detects depression early by capturing behavioral changes while continuously monitoring the behavior of the elderly in a smart home environment in which various sensors and their network are installed. Researchers in this field detect depression by recognizing changes in body shape such as a patient’s gait [13,14], head position [15], and thoracic kyphosis [16]. The method of detecting depression using the sensors is again divided into obtrusive and unobtrusive; the former has the problem of inconvenience and the latter suffers from low accuracy [5,17]. The other one detects depression early by analyzing behavior in cyberspace using various AI techniques [18]. However, unlike the young, the elderly in cyberspace are mostly passive users who enjoy surfing, watching, and enjoying, rather than active users who actively express their opinions and participate [19]. For this reason, it may be difficult to detect depression in the elderly at an early stage with a language model based on text mining and deep learning used in existing sentiment analysis. In addition, as mentioned previously, although sentiment changes and physical changes, which are signs of depression in the elderly, are highly correlated [16,20], there is very little research on them at the same time.

### 1.2. Motivation

This paper proposes a method for the earlier detection of depression in the elderly by considering both sentiment changes in cyberspace and physical changes. For this, the following assumptions are made: the elderly subject lives in a smart home that can monitor their daily activities, and his/her Internet activity logs and service usage records are all collectible. The first assumption is for sensor data related to physical changes in the elderly. The second assumption is for collecting sentiment change-related data required for early detection of depression. Even if sufficient data have been collected through this process, the following problems need to be addressed for the detection of depression in the elderly while considering both sentiment and physical changes. The first is data heterogeneity. Generally, sensor data are mostly signal data in numerical form, whereas log or service usage history data are a mixture of categorical and numerical data. Therefore, it is imperative to address the syntactic heterogeneity of data that are difficult to incorporate in matrix or tabular form due to the differences in the structures of data elements, as well as differences in terms of format and temporal characteristics. The second is the integration of changes in cyberspace and physical space related to the symptoms of depression in the elderly. To achieve this, it is necessary to use a modeling method for the cross-domain knowledge that connects them while maintaining the unique characteristics of the spaces. At this time, the cross-domain knowledge acts as a share point that can detect physical and sentiment-based changes. The third is the accurate capturing of minute changes in the physical activity and sentiment of elderly subjects. This requires a new modeling method to represent activity patterns and their minute changes.

### 1.3. Main Idea

To address the above issues, this paper proposes knowledge graph-based cyber–physical view-based activity pattern recognition for the early detection of depression, also known as KARE. At this time, a knowledge graph (KG) is defined as a multi-relational graph composed of entities as nodes and relations with labels to represent multidisciplinary domain knowledge. In the KARE framework, the KG plays key roles in providing cross-domain knowledge as well as resolving issues of grammatical and semantic heterogeneity in order to integrate the cyber and the physical world. In addition, it can flexibly express the patterns of different activities for each elderly subject. The KARE framework consists of two modules: a knowledge graph-based cyber–physical view representation (CPVR) module and a personalized activity pattern recognition (PAPR) module for detecting depression, the latter of which is based on anomaly detection.

CPVR module: The CPVR module is used to integrate views represented by cyberspace and the physical world as a graph structure using the KG. At this time, a view is defined as a set of essential information needed to describe an entity in cyberspace and the physical world, and can include concepts, terms, schema, features, and/or relationships.

PAPR module: This is a module that represents the activity patterns of the elderly in a graph and performs prediction model-based learning focused on the recognition of behavioral patterns using a Graph Neural Network (GNN) model. It undertakes the learning of normal activity patterns and detects depression earlier based on the learned model using the graphs of the activity patterns and the Graph Attention Network (GAT) model.

The rest of this paper is organized as follows. Section 2 summarizes the related works. Section 3 presents an illustrative example highlighting that the detection of depression should be carried out through the integration of cyberspace and the physical world. Section 4 describes the overall architecture of the KARE framework and its components. Section 5 demonstrates the performance of the proposed framework through several experiments. Finally, Section 6 presents the conclusions and future research.

## 2. Related Works

### 2.1. Knowledge Graph Alignment

Even though a Knowledge Graph (KG) contains a huge amount of information from various domains, it has several limitations that impede its widespread use [21,22]. First, the complexity and high dimensionality of the KG, which hinder the rapid information retrieval, query answering, and data integration, are caused by the knowledge representation of various domains with different schemas, vocabularies, and graph structures [23,24]. Second, the incompleteness and inconsistency of the KG is a very primitive phenomenon that can occur as the KG’s scale grows and the domain coverage of the KG widens. It may prevent the performance of reasoning using acquired knowledge as well as the discovery of new knowledge in the KG [25]. Lastly, the large-scale issue of the KG occurs due to the huge volume of the RDF data as well as the great diversity of the sources over multiple domains [26].

To resolve the limitations of the KG, knowledge graph alignment that interprets and predicts the relationship between two heterogeneous nodes is in the spotlight as a core technology. However, since the KG is represented as a triple form, it is difficult to align the KG with its original properties. Thus, knowledge graph embedding was usually adopted for the KG alignment methods. The KG embedding techniques can be divided into two categories [27,28]. One is translational distance models, and the other one is semantic matching models. The former use distance-based score functions. TransE [29], TransH [30], and TransD [27] are typical translational distance models. Since these models recognize graph patterns in KGs unidirectionally, the graph patterns that can be captured are limited. The latter model uses similarity-based scoring functions. RESCAL [31] and Semantic Matching Energy (SME) [32] are typical semantic matching models. Additionally, there is research focused on the embedding of RDF graphs into a continuous vector named RDF2Vec [33]. This method uses neural language models to embed RDF graphs into the vector. Before training the neural language model, the RDF graph should be transformed into sequences; graph walks and Weisfeiler–Lehman subtree RDF graph kernels [34] are mostly used for this translation.

Recently, with the development of the Graph Neural Network (GNN) model, KG alignment methods based on GNN representation learning have been proposed to recognize patterns in complex graph structures [35]. GNN representation learning is a method of representing KG nodes or graphs as low-dimension vectors that can effectively discriminate components using the predictive performance of the GNN model. At this time, the types of the GNN model utilized are the Graph Convolutional Network (GCN), GraphSAGE, and Graph Attention Network (GAT). The GCN is a model that is inspired by the Convolutional Neural Network (CNN); it receives a subset of the neighboring nodes of a node as an input and discovers low and dense dimensions that can differentiate nodes, and it is usually used in cross-lingual KG alignment [36,37,38,39]. GraphSAGE minimizes information loss by concatenating vectors of neighbors rather than summing them into a single value in the process of neighbor aggregation [40,41]. GAT utilizes the concept of attention to individually deal with the importance of neighbor nodes or relations [21,42,43,44,45,46,47]. Since each model has different characteristics and advantages, suitable models for KG alignment differ depending on the components and the topological structure of the KG.

### 2.2. Graph-Based Anomaly Detection

Since the methods of anomaly detection based on the distribution or pattern of existing tabular data have fundamental limitations in terms of capturing the complex relational information of objects, graph-based anomaly detection (GAD) research has recently received great attention in relation to overcoming the limitations of anomaly detection in tabular data [48,49]. These GAD studies can be classified according to whether they focus on the complexities or dynamics of graphs. The GAD studies on the complexity of graphs attempt to obtain embedding vectors with low dimensions from complex-dimensional graphs (in particular, attribute networks) and use embedding vectors to perform anomaly detection. The GCN, which is relatively simple and has low computational complexity, is often used as the main model because it has to deal with graphs of complex dimensions [50,51]. However, since the GCN has structural limitations in the interpretation of complex and dynamic relationships in graphs, some anomaly detection studies that hybridize GCN and GAT using the concept of attention have been proposed [52,53]. Since the GAD for the attributes network assumes one complex graph in which graphs of various domains are mixed, they focus on finding a dimension that can interpret the complex dimension of the graph by hybridizing various models instead of using one GCN model [54]. In other words, it aims to train a comprehensive and generalized model that can detect various types of anomalies from various data sources and graphs.

The GAD studies in dynamic graphs focus on dealing with anomalies of time dependence. In particular, some anomalies that can only be found in long-term graphs are receiving great attention because they show a very latent pattern to be distinguished from normal patterns, but also impose a fatal risk to the system. The core process of these studies can be divided into two components. The first one involves the creation of a temporal graph that can represent the time dependency in the graph structure, and the second one relates to the design of a model that can capture temporal patterns. NetWalk proposed clique embedding and reservoir sampling to quickly capture dynamically changing graph dimensions [55]. AnomRank classifies the types of anomalies into structural and node-relational anomalies and proposes a method to detect anomalies in two ways [56]. In addition, GAD approaches that incorporate graph clustering or community detection methods have been proposed for dynamic graphs [57,58]. However, since these methods concentrate on the time features related to the anomaly, the features that distinguish the nodes are somewhat overlooked. Therefore, there are cases where the detection of anomalies largely depends on the topological structure of the graph, and it is difficult to capture the change of the node itself.

## 3. Illustrative Scenario

A smart home is a space where devices for the physical world, such as sensors and devices, are deployed and operated, and a space where services related to the cyber world such as social media services and over-the-top (OTT) services are executed. Mrs. Jane, a resident of a smart home in her 70 s, is an active senior who shares her opinions, emotions, and status with acquaintances through her SNS activities. She recently went to the hospital for insomnia and loss of appetite, but her symptoms did not improve because she did not recognize by herself that there were symptoms of depression. Generally, the elderly have difficulty recognizing signs that they are depressed, so they often complain of physical symptoms such as insomnia and loss of appetite rather than complaining of emotional changes [5]. As a result, it is hard to detect depression in the elderly in the early stages, so treatment time is often missed, leading to serious social problems such as suicide in the elderly. To solve the problem, it is necessary to recognize changes in the elderly in cyberspace as well as in the physical world. Suppose a smart home system can detect keywords such as “death” and “loneliness” from a person’s Facebook while also finding that the lights in their house are darker and more filthy than usual. Then, it could contribute to the early detection of depression. However, regarding the detection of depression in the elderly at the early stages, although the objects of the physical space are tightly connected to services within cyberspace, they are unfortunately implemented and operated separately because one type is sensor-oriented whereas the other is service-oriented (Figure 1 left pane).

A challenging issue concerns the connecting of depression-related objects between physical space and cyberspace while maintaining interoperability between them. This can be tackled using methods such as data schema, ontologies, and data exchange models [59]. However, these methods have the following limitations: inability to flexibly manage the schema of newly added or changing objects and difficulty in the interpretation of relationships between the objects that have different schema. We adopted a knowledge graph that contains various cross-domain knowledge to overcome the limitations. Since the entire KG is huge, we propose an intermediary knowledge graph called the cyber–physical view (CPV) that extracts and represents the core elements (called view) of the KG required for the early detection of depression (Figure 1 right pane). In addition, we propose a method to develop CPV.

## 4. Overall Architecture

As depicted in Figure 2, the KARE framework consists of two modules, namely a cyber–physical view representation (CPVR) module and personalized activity pattern Recognition (PAPR) module. The CPVR module integrates the views represented by cyberspace and the physical world as a graph structure using the KG. The PAPR module delivers the activity patterns to detect depression earlier. Lastly, the KARE framework detects depression earlier using the trained Graph Attention Network (GAT) model.

### 4.1. Cyber–Physical View Representation Module

The CPVR module develops intermediary KG that can integrate the elements of cyberspace and the physical world that are related to the detection of depression into a graph structure. The intermediary KG consists of elements of the KG that can connect cyberspace and the physical world. To identify the elements of the KG, the CPVR module performs entity alignment with the KG and the physical world and cyberspace, respectively. At this time, the entity alignment is performed between attributes of the data schema and vertices of the KG. Prior to discussing these issues in detail, the attribute-related data used in the entire KARE framework are defined as follows.

**Definition** **1.**
*Attribute-associated data*

$\left(A_{i}\right)$

*comprise the set of all data associated with the*

$i^{th}$

*attribute, such as its name, its relationships to other entities, and its own instances, in the data schema. They are simply represented as follows.*

(1)
$$A_{i}=\left\{e_{i},na_{i},\;I_{i}\right\},\;\;2\leq i\leq N\;$$

*where*

$e_{i}$

*is an entity name having*

$A_{i}$

*as an attribute,*

$na_{i}$

*is a name of*

$A_{i}$

*, and*

$I_{i}$

*is an instance vector for the*

$i^{th}$

*attribute.*


**Definition** **2.**
*Triplets of Vertex*

$\left(V_{j}\right)$

*is a set of triplets of the*

$j^{th}$

*vertex in the KG. It is simply represented as follows.*

(2)
$$V_{j}=\left\{v_{j},P_{j},O_{j}\right\},\;\;2\leq j\;$$

*where*

$v_{j}$

*is a URI of the*

$j^{th}$

*vertex,*

$P_{j}$

*is a predicate vector corresponding to*

$O_{j}$

*,*

$O_{j}$

*is an object vector (*

$\left|P_{j}\right|=\left|O_{j}\right|$

*),*

$P_{j}$

*has predicate labels as elements, and*

$O_{j}$

*has vertices or Literals as elements (*

$O_{j}\subset \{v_{j^{\prime }},Literals\;|\;\forall j^{\prime },j^{\prime }\in j,\;j^{\prime }\neq j\}$

*).*


#### 4.1.1. Attribute Representation Learning with 1D-CNN Model

Since the vertices of the KG are graph-natured data and the attributes of the data schema are tabular data, it is difficult to find the equivalent relationship between the attributes and the vertices with the conventional entity alignment methods using the topological structure of the graph. To overcome this difficulty, we propose a novel entity alignment method to discover the shared space of the attributes and the vertices using the CNN model. The critical issue of the novel entity alignment method relates to the means of generating the attributes and the vertices of the fixed dimension as inputs to the CNN model. Since the vertices can be extracted as vertices of embedding vectors by applying GCN-based node representation learning [60], we focused only on generating the attributes of embedding vectors. As with word representation, one-hot encoding must be performed for each attribute to create an embedding vector for the attributes. However, as mentioned in Definition 1, the attributes do not exist independently but are related to several elements, such as other entities, their instances, etc. Thus, the process is required to generate one-hot encoding vectors for the attributes. In this study, we devised attribute representation learning with 1D-CNN by borrowing the approach of word representation learning. In other words, just as word representation learning extracts embedding vectors by capturing the relationship between the center word and neighbor words, the proposed attribute representation learning is a method of extracting embedding vectors by identifying the relationship between the center attribute and its related neighbor attributes. In the proposed method, the most important thing is to resolve the structural differences between the words and the attributes. Hereinafter, the solution will be described in detail.


**Step 1: Generation of the attribute sequence**


Generally, 1D-CNN sequentially receives a set of center words and neighbor words as input data according to a specific word sequence. However, the attribute-associated data are just a chunk of information, and there is no sequence between them. To generate the sequences of the neighbor attribute, two-phased sub-sampling is performed. The first-phase sub-sampling is performed to randomly select a set of the instance vectors of neighbor attributes of a specific attribute called the center attribute. The sampling procedure is as follows. Let one of $A_{i}$ be the central attribute $A_{c}$. At this time, $A_{c}$ has l neighbor attributes $A_{cr}\;$ that have $I_{cr}$ as an instance vector (r=1,2,…, l). In addition, $A_{cr^{\prime }}$ is a randomly selected $k^{1}$ size subset of $A_{cr}$ $(r^{\prime }=1,2,\ldots ,\;k^{1},\;k^{1}<l$) used to find the relationship between the central attribute and the neighbor attributes. For all $r^{\prime }$, $I_{cr^{\prime }}$ is randomly sorted, and an index p indicating the order of $I_{cr^{\prime }}$ is added according to the sorted order ($p=1,\ldots ,k^{1})\;$. Finally, the sequence vector of $A_{cr^{\prime }}$ is created as follows.
(3)$$nA_{c}=\left\{\left[\ldots ,I_{c(r^{’},p)},\ldots \right]\;|I_{c(r^{’},p)}\in A_{cr^{\prime }},\;\forall r^{\prime },r^{\prime }<l\right\}$$
where $k^{1}$ is the sampling size of first-phase sub-sampling.

The second-phase sub-sampling is performed for the selection of $k^{2}$ elements from $I_{c(r^{’},p)}$ (∀ p). Let $X_{c(r^{’},p)}$ be a part of $I_{c(r^{’},p)}\;$ with size $k^{2}$. Thus, the result of the second-phase sub-sampling—in other words, a sequence of the attribute $A_{c}$—is represented as follows.
(4)$$AS_{c}=\left\{\ldots ,X_{c(r^{’},p),}\ldots \;|\forall r^{\prime },r^{\prime }\leq k^{1}\right\}$$
where $k^{2}$ is the sampling size ($\left|X_{c(r^{’},p)}\right|=k^{2}$).

For each i, multiple random sampling is performed as the $A_{i}$ central attribute and the results are used to generate a series of sequences $AS_{i}^{s}\;\left(s\geq 2\right).$


**Step 2: Vector transformation**


Finally, to represent the attribute’s information as an initial input vector, one-hot-encoding is performed using the instance vector containing the most attribute information. As a result, $k^{2}\times N$ can be obtained as the input matrix even though only 1×N initial input vector is needed. Since the matrix cannot be used as input data for the 1D-CNN, vector transformation is performed as follows. First, 1×N empty vector of $k^{2}$ is created, similar to the approach used in one-hot encoding. At this time, N  is the number of attributes. Using $AS_{i}^{s}$ and its instance vector $X_{i(r^{’},p)}^{s}$, the related neighbor $r^{\prime }$ is identified and $X_{i(r^{’},p)}^{s}$ is assigned into the $r^{’th}$ column of the empty vector as a value. All other columns of the empty vector except $r^{\prime }$ are assigned dummy vectors as inputs. This process is repeated in sequence according to the p value. Finally, a matrix of encoded attribute sequence $eAS_{i}^{s}$ ($k^{1}\times N$) is obtained.

However, 1D-CNN uses only a single value as an input, not a matrix. Therefore, the matrix is converted to a single value through flattening and concatenation. As a result, matrix $eAS_{i}^{s}$ is converted into $fAS_{i}^{s}$ ($k^{1}\times (N\times k^{2}$)), which has a single value as an entry. The $fAS_{i}^{s}$ is an input sequence for the 1D-CNN training and the filter used for the training process is as follows.
(5)$$filter_{fl}^{h}=f\left(w\cdot fAS_{i}^{s}\left[fl:fl+h\right]+b\right)$$
where fl is the filter index, f is the activation function (relu),…,  is the dot product, $fAS_{i}^{s}\left[fl:fl+h\right]$ is a sub-matrix of $fAS_{i}^{s}$, h is the filter size, and b is the bias.

In addition, the loss function based on cross-entropy is as follows.
(6)$$loss=-\underset{i}{\sum }\underset{\iota }{\sum }y_{\iota }^{s}\;\log\left(p_{\iota }^{s}\right)$$
where $y_{\iota }^{s}$ is the $\iota^{th}$ true label of $A_{i}$ ($y_{\iota }^{s}\in I_{i}$), and $p_{\iota }^{s}$ is the $\iota^{th}$ prediction value.

The overall procedure for attribute representation learning with the 1D-CNN model is depicted in Figure 3. Finally, the attributes of the embedding vectors ($Eeb_{i})$ are obtained for all attributes.

#### 4.1.2. Entity Alignment with Embedding Vectors and CNN

In the next step, using the attributes and vertices of embedding vectors, entity alignment between the attributes and the vertices is performed. To achieve this, descriptive information such as the name or title of the entities expressed in the ‘rdf:label’ is absolutely necessary. However, the vertices of the embedding vectors contain descriptive information, whereas the attributes of the embedding vector contain only information on the relationship between the attributes based on the distribution of instances. Thus, to determine whether two vectors are equivalent or not, the attributes of the embedding vectors ($Eeb_{i}$) are first concatenated based on the name of the entity ($e_{i})\;$ and the attributes ($na_{i}).$ Next, the embedding vectors of the pair of the attribute and the vertex that has an equivalence relationship are concatenated and annotated with the label ‘1.’ This process is repeated for all attributes and vertex pairs that have a relationship of equivalence. Furthermore, the pairs of the attributes and the vertices that are not equivalent are randomly selected. After concatenating their embedding vectors, they are annotated with ‘0.’ Finally, CNN is performed using the concatenated and annotated vectors as inputs. The overall procedure of the entity alignment is summarized in Figure 4.

The trained CNN model is used to predict the equivalence of all pairs of the attribute and the vertex. When a pair of attributes and vertices that have an equivalence relationship is found, a triplet with the predicate ‘owl:sameAs’ is created and added to KG. In this case, it is assumed that all attributes can be identified by URI. Integrated knowledge graphs on which entity alignment has been performed are represented as follows.

**Definition** **3.**
*An integrated knowledge graph*

(IKG)

*is a set of triplets (subject, predicate, and object) including inferred equivalence relations.*

(7)
$$IKG=\left\{\ldots \left(s_{q},p_{q},o_{q}\right)\ldots \right\},\;\;2\leq q$$

*where*

$s_{q}$

*is a URI of the attribute or the vertex on the subject of the triplet,*

$p_{q}$

*is a predicate label on the predicate of the triplet (*

$‘owl:sameAS’\in \{p_{p}|\forall p\}$

*), and*

$o_{q}$

*is a URI of the attribute or vertex or*

Literals

*on the object of the triplet.*


#### 4.1.3. View Selection Based on Feature Selection Method

IKG is a vast graph containing various types of information such as concepts, terms, observations, and relations such as the equivalence of the attributes and the vertices. However, as already known, the GNN model restricts the structure and size of the input graph, i.e., the number of hops or nodes, due to inherent limitations. Thus, the large number of concepts and relationships that IKG has can be non-informative and noisy in terms of recognizing the graph patterns needed for the detection of depression. To remove the noise, we propose a method for extracting a sub-graph of the IKG, known as the cyber–physical view (CPV), composed of informative and representative nodes and relations that are needed to recognize the graph patterns that link cyberspace and the physical world. The CPV is represented in triplets as follows:(8)$$CPV=\left\{\ldots \left(s_{q^{\prime }},p_{q^{\prime }},o_{q^{\prime }}\right)\ldots \right\},CPV\subset IKG,2\leq q’$$

To devise a means of extracting the CPV from the IKG, we borrowed the feature selection method that finds the most informative and smallest subset of the original feature set. Applying the existing feature selection method requires prior knowledge such as label information or specific distance metrics. However, it is not possible to assume prior knowledge of IKG’s views. Therefore, this paper proposes two methods: unsupervised pseudo-label generation using GNN representation learning and wrapper-based view selection. At this time, the first one is used to generate embedding vectors of a full-size sub-graph connected to an arbitrary node and its partial sub-graphs based on GNN representation learning. After calculating the distance between the full-size embedding vector and the embedding vectors of the partial sub-graphs, the partial sub-graphs with the smallest difference to the full-size embedding vector in terms of size are selected as a view. The result is a CPV, which is a combination of the graphs made up of only informative triplets.

### 4.2. Personalized Activity Pattern Recognition

The behavioral patterns of elderly subjects living in smart homes are very different depending on their individual interests, habits, and health status. In addition, since the behavioral patterns are time-dependent, even the same behavior can have different time intervals. For the early detection of depression based on recognizing minute changes in the normal activity patterns, an informative and flexible graph of time features as well as the GNN model are needed. However, since the CPV contains all kinds of informative and representative nodes and relations that link cyberspace and the physical world, it is necessary to identify those related to activity patterns. Three steps are involved in the discovery of the activity patterns: finding activities of attributes via data-centric approaches, generating activity graphs using CPV triplet augmentation associated with the activities of attributes, and arranging activity graphs sequentially. As a result of the discovery of the activity patterns, activity pattern graphs are acquired and used as inputs to learn the GAT that can capture the normal activity patterns. Finally, early detection of depression is performed with the trained GAT performs.

#### 4.2.1. Generation of the Activity Pattern Graphs

It is difficult to find the activities directly in CPV due to the complexity of the graph structures. Thus, for this stage in the process, we decided to focus on finding the activities of attributes using data; in other words, instances. At this time, it is important to find the timestamp and the time intervals of each activity, but it is assumed that they are given because these aspects are beyond the scope of this paper. The finding of the activities of attributes via a data-centric approach is performed with the time-window drift and Gaussian Mixture Model (GMM). Prior to a detailed step-by-step description that follows, Figure 5 schematically illustrates the generation procedure of activity pattern graphs.


**Step 1: Finding the activities of the attributes**


As shown in Figure 5, there are instances for every attribute (e.g., $A_{traffic}$ and $A_{rotation}$) along the timeline. However, since not all instances of the attributes are related to the activities, it is necessary to find only instances that are related to the activities. To achieve this, the instances are sliced to the size of the time window with a given hyperparameter. As a result, an attribute-specific dataset consisting of a series of instances with a time interval w is created. For all instances $I_{i}$ (for all i) with time interval w, their start time and end time are assigned and then sorted in chronological order.
(9)$$I_{i}^{ti}=\left\{x_{iv}^{ti},s_{i}^{ti},e_{i}^{ti}|x_{i}^{ti}\in I_{i},v=1,2,\ldots \right\},\;1\leq ti$$
where $x_{iv}^{ti}$ is the $v^{th}$ instance of $I_{i}$ in the $ti^{th}$ time interval $\left[s^{ti},e^{ti}\right]$, and $s^{ti}$ and $e^{ti}$ are the start time and the end time of the $ti^{th}$ time interval ($s^{ti}<e^{ti},\;\left|s^{ti}-e^{ti}\right|=w$), respectively.

Equation (10) shows the probability distribution of the $I_{i}^{ti}$ value estimated using GMM and the expectation-maximization (EM) algorithm.
(10)$$p\left(x_{i}^{ti}\right)=\sum_{g=1}^{G}w_{g}\times N\left(x_{i}^{ti};\mu_{g},\sigma_{g}^{2}\;\right)$$
where $w_{g}$ is an EM-based estimated probability that the $g^{th}$ Gaussian distribution is selected, and $\mu_{g}$ and $\sigma_{g}^{2}$ are the EM-based estimated mean and standard deviation, respectively. G is the number of Gaussian distributions.

To determine whether the instances, which are the elements of the attribute-specific dataset, contains information on the activities, $p\left(x_{i}^{ti}\right)$ is compared with the distribution of instances in the sleep state called $p\left(x_{i}^{sleep}\right)$, which does not carry any activities. At this time, it is assumed that $p\left(x_{i}^{sleep}\right)$ has already been generated using the sleep dataset and Equation (10). Finally, the distance between the two probability distributions is calculated using the Kullback–Leibler distances Equation (11).
(11)$$KL_{i}^{tl}=\int_{-\infty }^{\infty }p\left(x_{i}^{non}\right)log\frac{p\left(x_{i}^{non}\right)}{p\left(x_{i}^{ti}\right)}dx_{i}$$

If $KL_{i}^{tl}\;$ is greater than the threshold δ, the $ti^{th}\;$ activity of $A_{i}$ is assigned as an element of $AA_{i}^{ti^{*}}$($\left|ti^{*}\right|<\left|ti\right|$). For all ti and i, the above process is repeated and $AA_{i}=\{AA_{i}^{ti^{*}}|\forall ti^{*}\}$ and $AA=\{AA_{i}|\forall i\}$ are generated.


**Step 2: Generation of activity graphs using the CPV**


$AA_{r}$ is randomly selected from the set AA (r∈i), and subsequently, the node of $AA_{r}$ sharing the index with $AA_{r}$ is discovered. A sub-graph composed of $A_{r}$ and its neighbor triplets is extracted from the CPV. By using the extracted sub-graphs and the $AA_{i}^{ti^{*}}$, the activity graphs $AG_{i}^{ti^{*}}\;$ of the $i^{th}$ attribute are created as follows:(12)$$AG_{i}^{ti^{*}}=\{AA_{i}^{ti^{*}},\left\{\ldots ,\left(s_{i}^{ti^{*}},p_{i}^{ti^{*}},o_{i}^{ti^{*}}\right),\ldots \right\}|s_{i}^{ti^{*}},p_{i}^{ti^{*}},o_{i}^{ti^{*}}\in CPV\}$$

At this time, although $AG_{i}^{ti^{*}}$ generated from $A_{i}$ is a graph of the same structure, it is possible to classify it according to $AA_{i}^{ti^{*}}$. The above process is repeated for all $ti^{*}$ and i values to obtain the sets $AG_{i}=\{AG_{i}^{ti^{*}}|\forall ti^{*}\}$ and $AG=\{AG_{i}|\forall i\}$.


**Step 3: Arrangement of activity graphs to represent the activity pattern graphs**


To represent the activity patterns in the graphs, the relationship between $AG_{i}^{ti^{*}}$ is identified and serialized. At this time, the relationship is of two types: a time series relationship and a path relationship. The former is the back-and-forth relationship between the $AG_{i}^{ti^{*}}$ elements of the set $AG_{i}$. This relationship is established if the time difference between arbitrary values of $AG_{i}^{ti^{*}=a}$ and $AG_{i}^{ti^{*}=b}$ is less than the size w ($\left|e^{tl^{*}=a}-s^{tl^{*}=b}\right|<w$). The latter establishes a pair of activity graphs $(AG_{i=a}^{ti^{*}=c},AG_{i=b}^{ti^{*}=c}$) where $AG_{i=a}$ and $AG_{i=b}\;$ share $tl^{*}$ but are elements of different sets. It is created by extracting triplets between $A_{i=a}$ and $A_{i=b}$ in the CPV. Finally, the activity patterns in the graphs of $APG_{i}^{ti^{*}}$ are created: ($\left|APG_{i}^{ti^{*}}\right|\geq \left|AG_{i}^{ti^{*}}\right|$).

#### 4.2.2. GAT-Based Activity Pattern Recognition

The Graph Convolutional Network (GCN), GraphSAGE, and Graph Attention Network (GAT) are mainly used as GNN models to capture graph patterns [61]. Each method has the following pros and cons. The GCN has the advantage of being able to preserve the node or subgraph information of very complex graph topologies, but it is difficult to capture the relationship between them. GraphSAGE can preserve information on the relationships between the nodes and their neighbors more effectively than the GCN, but its generalization performance requires a large amount of data due to the concatenation process. Finally, the GAT has an advantage in terms of its generalization performance because it effectively preserves relational information between nodes and their neighbors with the concept of attention but has a high computational complexity. As mentioned previously, the input data $APG_{i}^{ti^{*}}$ to be used for training is a small graph related to the activities and a temporal graph focusing on the relationship between the nodes along the timeline of the activities. Since the GAT is suitable for such $APG_{i}^{ti^{*}}$ data, the GAT and $APG_{i}^{ti^{*}}$ are used to train a GAT model so that it can recognize normal activity patterns. By applying this model, minute changes that are different from normal activity patterns, that is, symptoms of depression, can be detected earlier.

## 5. Evaluation

In order to evaluate the superiority of the proposed KARE method, we conducted experiments and evaluations using the state-of-the-art comparative models with experimental graph datasets covering multidisciplinary domains. We used a system with an Intel(R) Xeon(R) CPU @ 2.30 GHz processor, a Teslar P100 graphics card, and 12 GB memory

### 5.1. Experimental Datasets

Experimental graph datasets collected from multiple domains such as the KG were required, and information such as the label text needed to be attached to the components of the graph. In addition, label information for anomalies was required to evaluate the anomaly detection performance. As the most suitable dataset for these requirements, we used the *redditLinks* dataset that graphs the relationship between the body of posts and the topics in multi-topic discussions on the Reddit platform [62].

The *redditLinks* dataset is the most suitable graph for the KG entity alignment and anomaly detection that we chose to experiment with, but the structure of the graph, especially the entity-to-entity connection relationship, is somewhat different. In general, the relationship between entities in the graph is established directly, but in the *redditLinks* dataset, the relationship between entities is composed via posts. We used the posts as a predicate to connect entities and generate triples. Additionally, on the *redditLinks* dataset, the title of the entity is attached, not the ID of the individual entity. After replacing all of these with IDs, we created relations of equivalence using the entity’s titles. The created equivalence relations were used as a training and test set for the KG alignment. Finally, the features or attributes of each entity on the *redditLinks* dataset were represented as sets of vectors of features rather than attached in a graph form. We disassembled vectors and reconstructed them in the form of a graph centered on all entities. Figure 6 shows the structure of the graph transformed from the *redditLinks*’s graph for our experiment. The statistics of the transformed dataset are summarized in Table 1.

### 5.2. Knowledge Graph Alignment

We proposed a method of KG entity alignment based on attribute representation learning. To evaluate the performance of the proposed method, five state-of-the-art models of KG entity alignment were selected. MtarnsE, TransD, RotatE, and ConvE do not utilize the relationships of attributes among the entities when embedding the entity. On the other hand, GCn-align uses the relationships of attributes to embed entities, similar to the KARE framework. MtarnsE, TransD, RotatE, and ConvE do not utilize the relationships of attributes among the entities’ relationships for the embedding of entities. On the other hand, GCN-align uses the relationship of attributes to find the dimensions of entities, similar to the KARE framework. Each model was trained by limiting it to the epoch 50, and the default parameters were adopted for the training models. In addition, since the scale of the experimental dataset was very large, it was sampled and tested. Table 2 shows the experimental results of the full dataset with the metrics of Hits@k, mean rank (MR), and mean reciprocal rank (MRR). Hits@k is the percentage of corrected alignment in the top k. MR and MRR are the averages of ranks and inverse ranks for test sets, respectively. Figure 7 shows the experimental results of the sampling datasets obtained for the five metrics.

**TransH** [30]: Translation-based model for a relation as a hyperplane.**TransD** [27]: Translation-based model to improve TransR/CTransR with representation of a named symbol object.**RotatE** [63]: Relational pattern-based KG embedding model including relationships of symmetry/antisymmetry, inversion, and composition.**ConvE** [36]: A multi-layer convolutional network model used to improve the performance of extracted feature sets from KG for link prediction.**GCN-align** [64]: A graph convolutional network (GCN)-based KG entity alignment model based on both the structural and attribute information of entities.

As a result of embedding in the full dataset, the proposed KARE model was found to be superior in all metrics. The MRR metric showed the best performance, and it can be seen that the KARE model did not perform erroneous alignment for unrelated equivalence relationships, that is, for relationships in which priority was considered. In other words, the accuracy and precision of the alignment of the KARE model showed the best performance compared to the state-of-the-art methods. KARE showed a rather low value for MR, but the other models fitted the majority equivalence relationship well; these values were found to be relatively high compared to that obtained for the KARE model.

As the sampling ratio changed, we conducted an experiment to compare the performance using five metrics and comparative models. The results of the experiment are shown in Table 3 and Figure 7.

As a result of the experiment, the proposed KARE model showed high accuracy at many of the sampling ratios. The Hit@k metric decreased as the sampling ratio increased, because the complexity of the equivalence relationship to be matched increased. On the other hand, the MR and MRR values moved in opposite directions, because in the case of MR, they increased when the same equivalence relationship was continuously met, and MRR fell because the wrong equivalence relationship must have been met. In the interval where the sampling ratio was low, the performance of most models deteriorated because the complexity of the sampling set increased faster than the information required to find the generalized embedding space. The model that showed the most similar performance to our method was the GCN-align model. Most of the KARE methods showed higher values than GCN-align, but in the last 50% of the sampling process, GCN-align showed significantly higher values. This was because the information about the hyperplane to be discovered by the GCN-align model exceeded the threshold. In addition, we found a rather low value in the first 50% of the sampling process because the information loss in the embedding process was not obtained in this 50% sampling process. However, taking into account the process of the entire dataset, as shown in Table 2, the proposed KARE method showed the best performance for the full dataset.

## 6. Conclusions and Further Research

In this paper, we proposed the use of the KARE framework, which enables the early detection of the depression in the elderly by integrating data from the physical world and the cyber world. To achieve this, we developed the KG and the attribute representation learning method for the alignment of the KG and attributes of the data schema. Since the integrated KG, which is obtained by alignment, contains many duplicate entities and unnecessary graph structures for the detection of depression, we utilized graph representation learning and wrapper-based feature selection to extract CPV from the graph, which was integrated using the unsupervised method. Lastly, we proposed a method to generate activity pattern graphs using GMM and KL divergence. Using these activity pattern graphs, the GAT model was trained for the detection of normal activity patterns, and the early detection of depression was performed. Since the proposed KARE framework integrates physical space and cyberspace to detect observable anomalies based on human behavior, it can be applied in various scenarios in the e-health care sector. In addition, if events are detected instead of anomalies, the KARE framework can be extended to recommender systems or decision support systems. When detecting anomalies in system components rather than humans, it can be applied to various applications within the context of intrusion detection and cyberattack defense, such as DDoS attacks.

However, this paper has several limitations. To perform alignment of the KG and the attributes of the data schema, we assumed the existence of relationships of equivalence between the KG and the data schema. In real world applications, it is hard to assume that the information on relationships of equivalence exists. Therefore, a method that detects relationships of equivalence using unsupervised methods is needed to align heterogeneous graphs. Moreover, our method depends on the numerical space of the instances to generate the activity graphs. Even though categorical data can be projected into numerical space, some non-time series data cannot be detected by embedding numerical space, which preserves the original information. The KNN or decision tree can be appropriate models to handle categorical and non-linear data and transform original data to the continuous space to find the best embedding space as a preprocessing model. Therefore, a more robust and general method is needed to recognize activity graphs.

Further research will be performed to overcome the limitations mentioned above. First, using natural language processing methods and graph representation learning, the GNN model will be developed to detect equivalence between heterogeneous graphs, not between KGs. Secondly, to generate activity graphs from attributes of the data schema, a multi-view CNN model will be developed and applied to deal with a wide variety of data. We will continue to conduct research on assistive AI technology that can detect depression in the elderly at an early stage and lead a safe life within smart homes.

## Figures and Tables

**Figure 1 sensors-22-01545-f001:**
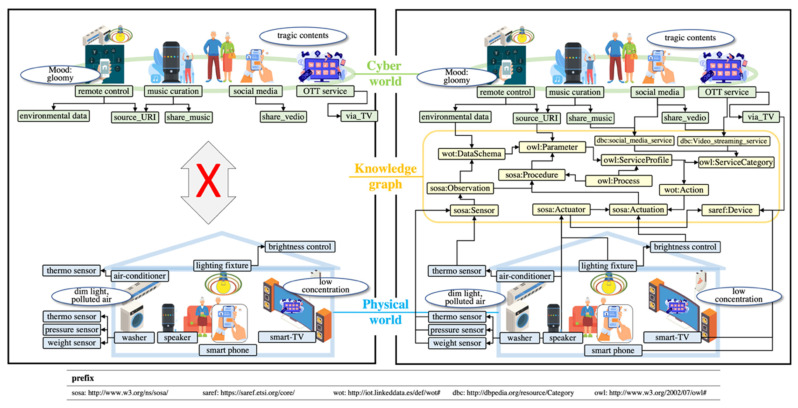
Illustrative example of a knowledge graph-based cyber–physical view (CPV).

**Figure 2 sensors-22-01545-f002:**
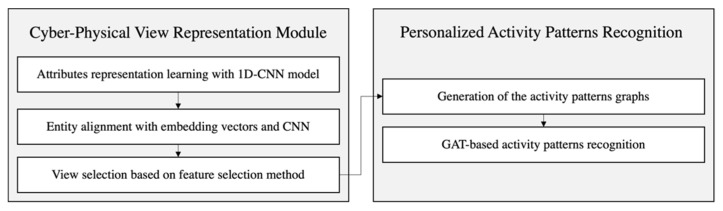
Overall structure of the KARE framework.

**Figure 3 sensors-22-01545-f003:**
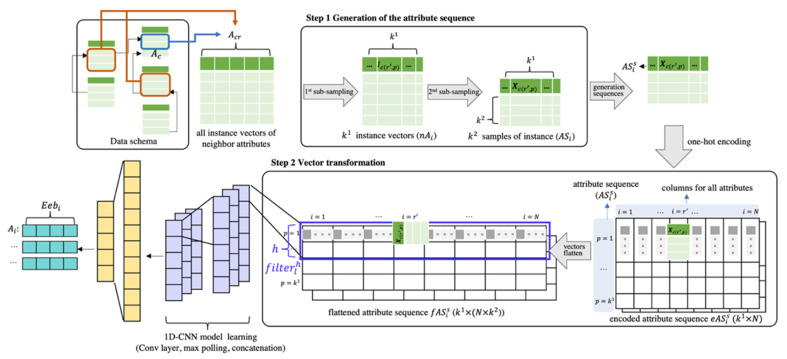
The proposed attribute representation learning with the 1D-CNN.

**Figure 4 sensors-22-01545-f004:**
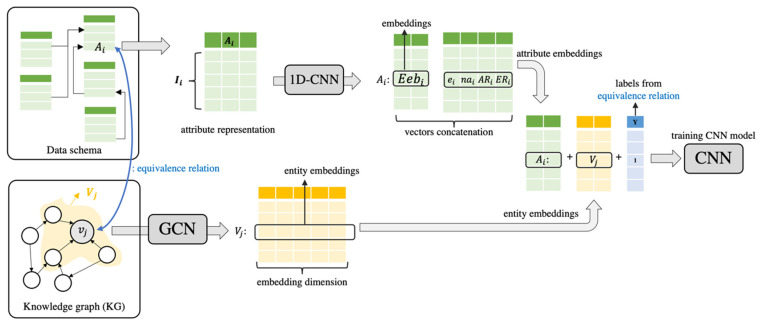
The entity alignment between attributes and vertices with the CNN model.

**Figure 5 sensors-22-01545-f005:**
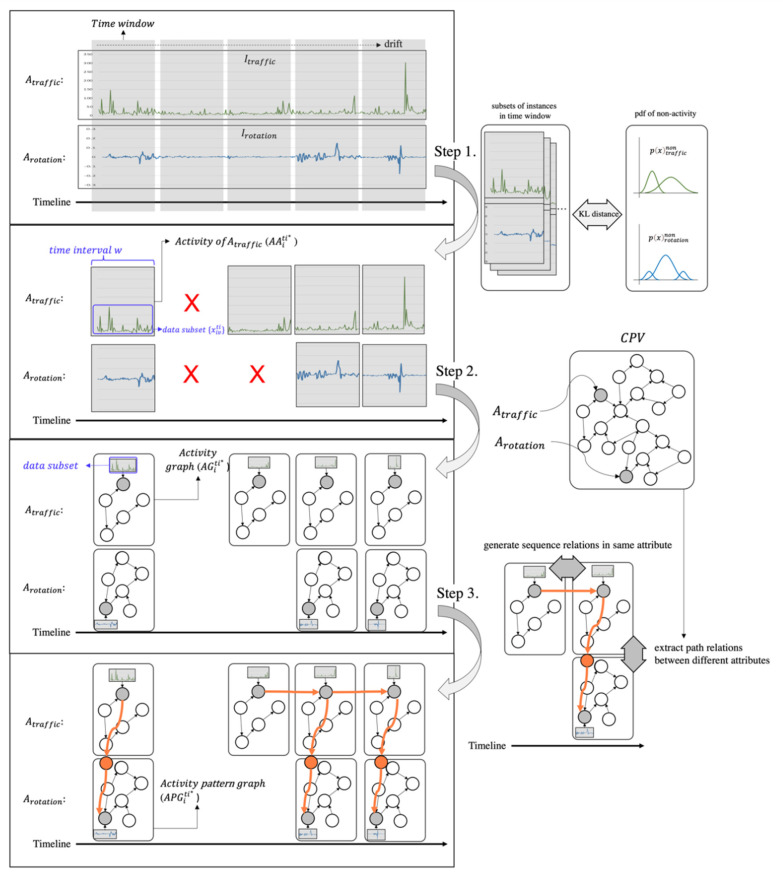
The generation procedure of activity patterns graphs.

**Figure 6 sensors-22-01545-f006:**
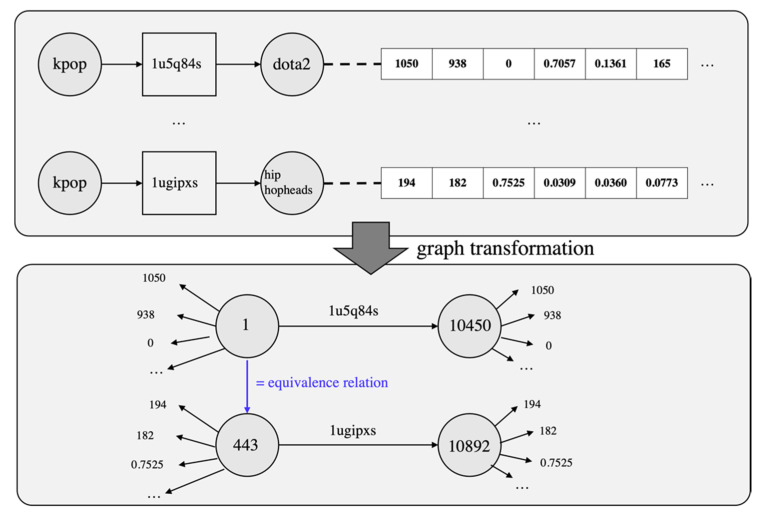
Graph transformation process to convert *redditLinks* graph to KG-like structure.

**Figure 7 sensors-22-01545-f007:**
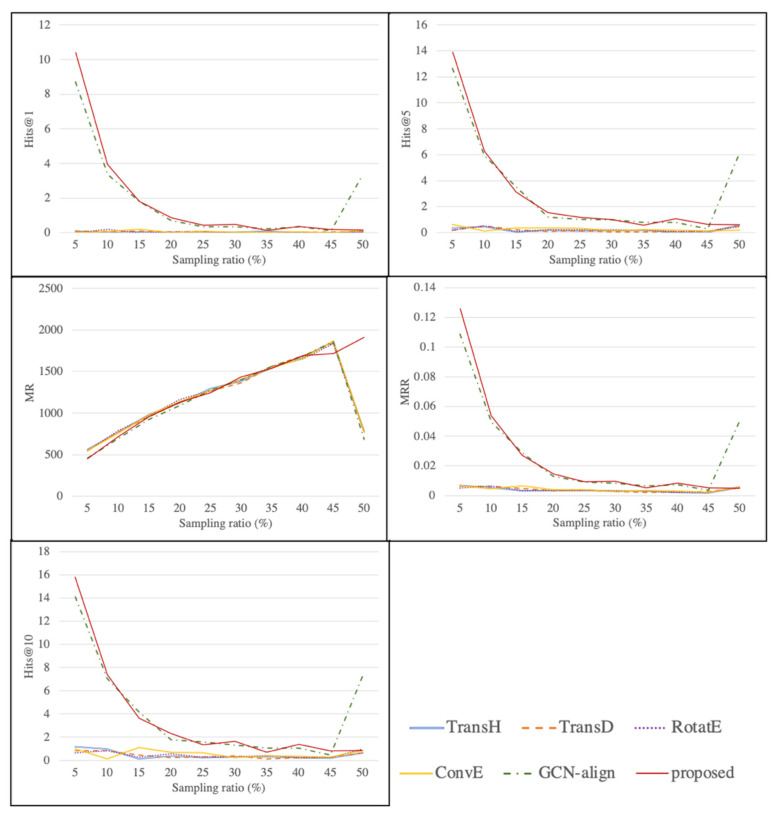
Comparison of knowledge graph alignment methods according to sampling ratio.

**Table 1 sensors-22-01545-t001:** The summary of the transformed dataset from *redditLinks*.

#Entities	#Entity Titles	#Entity Relations	#Predicates	#Attributes	#Attributes Relations	#Anomalies (%)
573,122	35,776	286,561	259,092	86	3,438,732	21,070 (7.93%)

**Table 2 sensors-22-01545-t002:** Results of knowledge graph alignment with the full dataset with 5 metrics.

Methods	Hits@1	Hits@5	Hits@10	MR	MRR
TransH	0	0.11	0.201	2744.106	0.001517
TransD	0.018	0.11	0.201	2759.899	0.001858
RotatE	0	0.073	0.146	**2724.465**	0.001502
ConvE	0.018	0.073	0.146	2759.12	0.001571
GCN-align	0.073	0.238	0.384	2763.467	0.002591
**KARE**	**0.091**	**0.328**	**0.492**	2790.59	**0.003123**

**Table 3 sensors-22-01545-t003:** Results of knowledge graph alignment with the sampling datasets.

**Methods**	**Hits@1**									
	5	10	15	20	25	30	35	40	45	50
TransH	0.091	0	0.052	0	0	0.036	0.097	0	0	0
TransD	0.091	0.065	0.103	0.044	0.077	0	0	0.03	0.054	0.065
RotatE	0	0.194	0	0.044	0.077	0.036	0.032	0	0	0.065
ConvE	0	0.065	0.207	0	0.077	0.036	0.032	0.059	0	0.129
GCN-align	8.719	3.355	1.808	0.709	0.348	0.358	**0.226**	**0.355**	0.136	**3.355**
**KARE**	**10.404**	**3.95**	**1.824**	**0.854**	**0.436**	**0.484**	0.129	**0.355**	**0.177**	0.158
**Methods**	**Hits@5**									
TransH	0.363	0.452	0.052	0.177	0.116	0.143	0.194	0.03	0.027	0.581
TransD	0.182	0.516	0.207	0.089	0.154	0.072	0.065	0.089	0.081	0.516
RotatE	0.182	0.516	0.103	0.222	0.232	0.215	0.194	0.089	0.109	0.452
ConvE	0.636	0.129	0.362	0.355	0.309	0.143	0.226	0.178	0.109	0.194
GCN-align	12.625	**5.935**	**3.512**	1.197	1.043	1.002	**0.806**	0.8	0.298	**6.129**
**KARE**	**13.901**	6.27	3.105	**1.537**	**1.188**	**1.003**	0.581	**1.066**	**0.648**	0.606
**Methods**	**Hits@10**									
TransH	1.181	0.968	0.103	0.399	0.232	0.322	0.29	0.237	0.19	0.645
TransD	0.817	0.839	0.465	0.222	0.309	0.394	0.129	0.237	0.271	0.645
RotatE	0.636	0.839	0.258	0.576	0.27	0.322	0.419	0.237	0.217	0.968
ConvE	0.999	0.129	1.085	0.665	0.657	0.215	0.387	0.326	0.271	0.839
GCN-align	14.078	7.097	**4.184**	1.773	**1.584**	1.324	**1.065**	1.066	0.461	**7.29**
**KARE**	**15.785**	**7.398**	3.647	**2.306**	1.347	**1.626**	0.71	**1.392**	**0.804**	0.843
**Methods**	**MR**									
TransH	563.444	768.733	981.909	1121.072	1296.727	1384.305	1548.724	1674.015	1860.349	778.442
TransD	551.669	774.206	953.977	1130.202	1265.188	**1362.231**	1560.129	1686.697	1837.794	774.777
RotatE	548.125	789.419	964.599	1163.451	1270.366	1407.159	1559.162	**1653.589**	1836.923	773.315
ConvE	547.085	759.823	978.21	1140.227	1271.151	1403.523	1544.462	1656.773	1865.935	765.801
GCN-align	458.357	**697.176**	**926.083**	**1094.491**	1272.667	1399.527	1564.627	1683.827	1858.828	**685.275**
**KARE**	**453.374**	718.921	960.947	1133.846	**1243.791**	1434.639	**1537.697**	1688.802	**1716.402**	1912.765
**Methods**	**MRR**									
TransH	0.006711	0.005399	0.003258	0.003317	0.357837	0.003011	0.003105	0.002225	0.001816	0.005183
TransD	0.006099	0.006153	0.004654	0.003295	0.003643	0.002698	0.002158	0.002255	0.002584	0.005378
RotatE	0.005174	0.006487	0.003308	0.003637	0.003468	0.003138	0.002955	0.002227	0.002078	0.005829
ConvE	0.006539	0.00447	0.006395	0.003954	0.003925	0.002785	0.003282	0.003082	0.002227	0.005568
GCN-align	0.108702	0.049645	**0.028628**	0.012819	0.009116	0.008219	**0.006439**	0.007301	0.003622	**0.049873**
**KARE**	**0.125774**	**0.053697**	0.027173	**0.014584**	**0.009398**	**0.009434**	0.005226	**0.008408**	**0.005283**	0.004791

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
