# Peer review of "Graph Representation Learning-Based Early Depression Detection Framework in Smart Home Environments"

_sensors, 2022, doi:10.3390/s22041545_

Round 1

Reviewer 1 Report

This paper proposes Knowledge graph-based cyber-physical vie (CPV)-based 13 Activity patterns Recognition for Early detection of depression, which is based on several machine learning methods. Five typical methods are compared based on the redditLinks dataset. Some comments:

From the results in Table 3 and Figure 7, error of GCN-align method increases when the sampling ratio is high. The authors need to explain this observation. Is that because of the network structure, like the number of layers and neurons, etc.? 

In addition, how to select hyperparameters for different neural networks, like multi-layer convolutional network or Graph convolutional networks?

In Figure 7, it is a bit confusing, why increasing the sampling ratio, the percentage of corrected alignment is decreasing? Please explain.

Author Response

Dear reviewer,

We are glad to receive the significant comments to improve the submitted paper. The revision is performed according to individual comments and the changes can be tracked using the function of MS word. The details of response and revision contents are summarized in the attached file. Please see the attachment.

Thank you again for your thoughtful consideration.

Sincerely,

Dr. Jongmo Kim (contact: [email protected] or [email protected] )

Department of Industrial Engineering

Sungkyunkwan University

300 Cheoncheon-dong, Jangan-Gu

Suwon, Korea

Tel: +82-0-3015-4072; Fax: +82-31-290-7610

Reviewer 2 Report

This manuscript Sensors-1560252 proposes KARE, a framework for early detection of depression in the elderly through a smart home environment. The model is constructed by the method based on graph learning, and the knowledge graph technology is used to realize the cross-domain knowledge integration of the cyberspace and the physical world. A 1D-CNN attribute representation learning method is designed to represent the attributes in the Cyber-Physical space. By establishing the entity alignment CNN model between attributes and nodes, the attribute entity alignment between cyberspace and physical time is realized, and finally integrated into KARE framework, abnormal behavior detection through KARE framework. The experimental results show that KARE has achieved better results on the whole in various evaluation indicators in the full datasets and the sampling datasets, which proves the effectiveness of this entity alignment method. My general impression of this paper is that it is generally well organized. It is a pleasure to review this work and I can recommend it for publication in the Sensors after minor revision. I respectfully recommend my comment below to the author.

(1) The English needs to be revised throughout. The authors should pay attention to the spelling and grammar throughout this work. I would only respectfully recommend that the authors perform this revision or seek the help of someone who can aid the authors.

(2) The reviewer suggest authors reduce the length of Introduction or add some secondary titles in Introduction for easier reading.

(3) (Page 2, Section I Introduction, Line 47) The reviewer suggests two references to this sentence that “Researchers in this field detect depression by recognizing changes in body shape changes such as a patient’s gait [9, 10], head position [1][2] {[1] DOI: 10.1109/TII.2022.3143605, [2] DOI: 10.1109/TMM.2021.3081873}, ….

(4) It is recommended that the authors rearrange and plan The 4 Overall Architecture to make the content more concise and clear, and some steps can be integrated together.

(5) (Page 2, Line 49) Please add some related references. The original statement that “The other one detects depression early by analyzing behavior in cyberspace using various AI techniques [14]+[1][2]. {[1] https://doi.org/10.1016/j.neucom.2020.12.090, [2] https://doi.org/10.1016/j.infrared.2021.103740}

(6) In the experimental part,the MR in Table 2 should not bold your own data, but should bold the smallest value in the MR column.

(7) (Page 3, Section I, 2.1 Knowledge Graph Alignment) The reviewer suggest to add some related work in the original statement “Even though Knowledge Graph (KG) [1][2] contains a huge amount of information of various domains, it has several limitations which impede its widespread use. {[1] DOI: 10.1109/TNNLS.2021.3055147, [2] DOI: 10.1109/TKDE.2020.3005952, }

(8) In the experimental part,there is no need to make Table 3 so redundant. For each evaluation index, you can select several columns from 5 to 50. The optimal data in each column should be marked bold.

(9) (Page 3, Section I, 2.1 Knowledge Graph Alignment) The reviewer suggest to add some related work in the sentence “Recently, with the development of the Graph Neural Network (GNN) model, KG alignment methods based on GNN representation learning have been proposed to recognize patterns in complex graph structures [1][2]. {[1] https://doi.org/10.1016/j.neucom.2021.10.050, [2] https://doi.org/10.1016/j.neucom.2020.07.137}

(10) The functions or applications of the KARE framework in terms of scalability can be briefly shown by some examples.

My overall impression of this manuscript is that it is in general well-organized. I would like to check and accept the revised manuscript. The work seems interesting and the technical contributions are solid.

Author Response

(The authors gave the same response as above.)

Reviewer 3 Report

The paper’s subject is interesting. Authors consider one of the possible problems which can be implemented in the framework of smart technologies. The paper presentation is well. The result and method are presented clearly. The background of the proposed method is the application of CNN. But the efficiency of CNN is depended on the number of samples/instances. Could other approaches be used instead NN, for example, as a decision tree or KNN? Could you consider in the introduction/section 2 or conclusion other approaches?

As the important aspects for the development of the method should consider: the application of fuzzy representation of the initial data and the influence of human factor in the concept of reliability analysis. Could you consider this possibility based, for example, the publications as:

Podofillini, L.; Dang, V.N.; Zio, E.; Baraldi, P.; Librizzi, M. Using Expert Models in Human Reliability Analysis – A Dependence Assessment Method Based on Fuzzy Logic, Risk Analysis, 30(8), 2010, 1277-1297.

Zaitseva, E.; Levashenko, V.; Rabcan, J.; Krsak, E. Application of the Structure Function in the Evaluation of the Human Factor in Healthcare. Symmetry 2020, 12, 93.

Zaitseva, E.; Levashenko, V. Construction of a Reliability Structure Function Based on Uncertain Data, IEEE Trans. on Reliab. 2016, 65(4), 1710–1723.

Baraldi, P.; Podofillini, L.; Mkrtchyan, L.; Zio, E.; Dang V.N. Comparing the treatment of uncertainty in Bayesian networks and fuzzy expert systems used for a human reliability analysis application. Reliab. Eng. Syst. Saf. 2015, 138, 176–193.

Author Response

(The authors gave the same response as above.)
